# Inhibitory effects of polysaccharides from Korean ginseng berries on LPS-induced RAW264.7 macrophages

**Weerawan Rod-in[1,2,3], Utoomporn Surayot[4], SangGuan You[3], Woo Jung Park[3]\***

**1** Department of Agricultural Science, Faculty of Agriculture Natural Resources and Environment, Naresuan University, Phitsanulok, Thailand, **2** Center of Excellence in Research for Agricultural Biotechnology, Faculty of Agriculture, Natural Resources and Environment, Naresuan University, Phitsanulok, Thailand, **3** Department of Marine Bio Food Science, Gangneung-Wonju National University, Gangneung, Gangwon, Korea, **4** College of Maritime Studies and Management, Chiang Mai University, Samut Sakhon, Thailand

* pwj0505@gwnu.ac.kr

**Data Availability Statement:** All relevant data are within the paper and its Supporting Information files.

**Funding:** This research was supported by Basic Science Research Program through the National

## Abstract

Polysaccharides isolated from Korean ginseng berries (GBPs) have shown beneficial effects such as immunomodulatory, anti-inflammatory, anti-cancer, and anti-diabetic properties. However, little is known about anti-inflammatory effects of GBPs. Thus, the purpose of this study was to investigate anti-inflammatory properties of four fractions of GBPs, namely GBP-C, GBP-F1, GBP-F2, and GBP-F3, in macrophages. Their toxicities and effects on NO production in RAW264.7 cells were assessed by culturing cells with various concentrations of GBPs and stimulating cells with LPS. Furthermore, expression levels of inflammatory mediators, cytokines, cell surface molecules, and immune signaling pathways were evaluated in LPS-stimulated macrophages using different fractions of GBPs at 450 µg/mL. These GBPs activated LPS-stimulated RAW264.7 cells to significantly reduce NO production. They suppressed the expression of mRNA and cell surface molecules via MAPK and NF-κB pathways. Collectively, results revealed that all four GBP fractions showed anti-inflammatory effects, with GBP-F1 having a more efficient anti-inflammatory effect than GBP-C, GBP-F2, and GBP-F3. The structure of GBP-F1 mainly consists of 1 → 3)- Ara*f*, 1 → 4)- Glc*p*, and 1 → 6)–Gal*p* glycosidic linkages. These results demonstrate that GBPs can be employed as alternative natural sources of anti-inflammatory agents.

## Introduction

Inflammation is a highly complex defense mechanism against outside harm or tissue damage [1]. A variety of stimuli can cause inflammation, including pathogens, noxious chemical and mechanical agents, and autoimmune reactions [2]. Macrophages are immune cells that play an important role in immunity. They can trigger an effective immune response by directly counteracting harmful stimuli such as lipopolysaccharide (LPS), muramyldipeptide (MDP), interferon-γ (IFN-γ), and granulocyte-macrophage colony-stimulating factor (GM-CSF) [2, 3]. In inflammation, major functions of macrophages are to present antigen, to phagocytose, and to

Research Foundation of Korea (NRF) funded by the Ministry of Education (RS-2023-00248832). This research project was also supported by the University Emphasis Research Institute Support Program (No.2018R1A61A03023584), which is funded by National Research Foundation of Korea.

**Competing interests:** The authors have declared that no competing interests exist.

modulate immunity through various cytokines and growth factors [2]. Additionally, macrophages could act as antigen-presenting cells (APCs) and modulate adaptive immunity by interacting with T lymphocytes [4]. It has been found that macrophages can modulate LPS uptake and signaling following activation by releasing accessory molecules like cluster of differentiation (CD) 14 and toll-like receptors 4 (TLR4) [6, 7]. Macrophages can also produce co-stimulatory molecules such as CD40, CD80, and CD86 that can promote sustained stimulatory interactions of T cells [5, 6].

LPS is an important element of Gram-negative bacteria that can activate many cellular signals in macrophages. It is widely used for inducing inflammation in macrophages [1, 7, 8]. As a result of LPS activation, macrophages can release inflammatory mediators such as nitric oxide (NO) and prostaglandin $E_2$ ($PGE_2$) through inducible nitric oxide synthase (iNOS) and cyclooxygenase-2 (COX-2), respectively. They also produce pro-inflammatory cytokines such as interleukin (IL)-1β, IL-6, and tumor necrosis factor (TNF)-α [9–12]. In addition to regulating the expression of these inflammatory mediators, the nuclear factor-kappa B (NF-κB) p65 subunit and mitogen-activated protein kinases (MAPK) also contribute to anti-inflammatory effects [10, 13, 14]. RAW264.7 cells are murine macrophages developed from BALB/c mice infected with Abelson leukemia virus (A-MuLV) [15]. They are useful models for examining immune activity of macrophages. Anti-inflammatory properties of polysaccharides have been demonstrated in RAW26.7 macrophages responding to LPS-induced inflammation, secreting inflammatory modulators and cytokines and activating NF-κB and MAPK signaling pathways [14, 16–18].

Korean ginseng (*Panax ginseng* C. A. Mey.), a member of the Araliaceae family, has been extensively utilized in traditional medicine to treat stress, heart failure, hypertension, and diabetes [19, 20]. Previous studies have shown that ginseng berries are rich in several ginsenosides (ginsenoside Re, Rc, Rg1, Rb1, Rb2, Rd) [21–23] as well as alkaloids [24], triterpene saponins [25], phenolic compounds [12], peptides [26], and polysaccharides [27, 28]. These components exhibit a number of pharmacological actions, including anticancer, anti-hyperglycemic, antioxidative, anti-inflammatory, and anti-aging properties [20]. In particular, polysaccharides are main active constituents of leaves, roots, and fruits of Korean ginseng known to contain primarily monosaccharides of galacturonic acid (GalA), galactose, arabinose, glucose, rhamnose, and xylose [29]. Ginseng berry polysaccharides can trigger immune responses such as immunomodulatory [30], immunostimulatory [31–33], anti-inflammatory [34], immunosenescence [28], anti-cancer [27, 31, 34], and anti-hyperglycemic [35] effects. Korean ginseng berries have anti-inflammatory effects of several active components against LPS-induced macrophages, which can decrease the production of NO, $PGE_2$, and some cytokines (IL-6 and TNF-α) [21, 26, 36, 37]. Anti-inflammatory effects of acidic polysaccharides in LPS-induced RAW264.7 cells derived from North American ginseng roots (*P. quinquefolius* L.) have been reported, including reductions of NO, IL-1β, IL-6, and TNF-α production [38]. *P. notoginseng* extracts can also inhibit LPS-induced TNF-α and IL-6 production, expression of *COX-2*, *IL-1β*, CD40, and CD86 in RAW264.7 cells [5]. Polysaccharides isolated from ginseng berry and fractionated polysaccharides can exert anti-inflammatory effects on malignant cells, human HT-29, and HCT-116 [34].

A previous study has identified crude and three fractionated polysaccharides isolated from Korean ginseng berries (GBPs: GBP-C, GBP-F1, GBP-F2, and GBP-F3) by an ion-exchange chromatography. In the analysis of these polysaccharides, gas chromatography-mass spectrometry (GC-MS) has been used to evaluate monosaccharides such as rhamnose, arabinose, mannose, glucose, and galactose [32]. Four fractions of polysaccharides have been shown to possess immune-enhancing activities by increasing the expression of *iNOS*, *COX-2*, *TNF-α*, *IL-1β*, and *IL-6* via NF-κB and MAPK signaling pathways [32]. GBP-C can also boost immune

function in mice [33]. However, GBPs have not been explored for their anti-inflammatory properties. Thus, the objective of the current study was to determine anti-inflammatory effects of GBPs in RAW246.7 cells treated with LPS and probable mechanisms involved in such effects.

## Materials and methods

### Samples

In accordance with our previous research, we obtained Korean ginseng berry polysaccharides (GBPs) consisting of crude polysaccharides (GBP-C) and three polysaccharide fractions (GBP-F1, GBP-F2, and GBP-F3) [32]. Molecular weights of GBP-C, GBP-F1, GBP-F2, and GBP-F3 were 328.4 and $54.2 \times 10^3$ g/mol, 115.7 and $16.0 \times 10^3$ g/mol, 107.7 and $40.0 \times 10^3$ g/mol, and 251.3 and $47.3 \times 10^3$ g/mol, respectively, analyzed by the HPSEC-UVMALLS- RI system [32]. All GBPs were dissolved in distilled water to evaluate their effects on macrophage cells.

### Cell culture and sample treatments

RAW264.7 cells were obtained from Koran Cell Line Bank (KCLB, RRID: CVCL_0493). These cells were cultured in RPMI-1640 medium (Gibco™, USA) supplemented with 10% fetal bovine serum (FBS, Welgene, Korea) and 1% penicillin/streptomycin (PS, Welgene, Korea) at 37˚C in a humidified incubator with 5% $CO_2$.

GBPs were diluted in RPMI-1640 medium (Gibco™, USA) supplemented with 1% FBS and 1% PS to different concentrations (50, 100, 250, and 450 μg/mL). For treatment groups, GBPs were added to cells at different concentrations. The negative control group was incubated with RPMI medium only. The positive control group was incubated with LPS only at a final concentration of 1 μg/mL. After 1 h of incubation, LPS at a final concentration of 1 μg/mL (Sigma-Aldrich, USA) was added to cells in groups treated with GBPs followed by incubation for another 24 h before performing assays to determine inhibitory effects of GBPs on inflammation.

### Determination of cytotoxicity and NO production

The culture medium of treated cells was collected after LPS stimulation to measure NO generation using Griess reagent (Promega, USA). Briefly, 100 μL of the culture medium was mixed with 50 μL of Griess reagent A (1% sulfanilamide in 5% phosphoric acid) and incubated at room temperature for 5 min. After that, 50 μL of Griess reagent B (0.1% *N*-1-napthylethylene-diamine dihydrochloride in water) was added followed by incubation for another 5 min. The absorbance of the solution was measured at 540 nm using a microplate reader (EL-800; BioTek Instruments, USA). An EZ-Cytox Cell Viability Assay Kit (Daeil Lab Service, Korea) was used to test the cytotoxicity of GBPs. The supernatant was discarded. The EZ-Cytox reagent was added to cells followed by incubation at 37˚C for 1 h. Absorbance at 450 nm was then measured using a microplate reader.

### Real-time PCR analysis

TRI reagent® (Molecular Research Center Inc., USA) was used to extract total RNA from cells. Reverse transcription of RNA to cDNA was performed with a High-capacity cDNA Reverse Transcription Kit (Applied Biosystems, USA). Real-time PCR was carried out to determine expression levels of genes (namely *iNOS*, *COX-2*, *IL-1β*, *IL-6*, *TNF-α*, and *β-actin*) using TB Green® Premix Ex Taq™ II (Takara Bio Inc., Japan) with a QuantStudio™ 3 FlexReal-Time PCR System (Thermo Fisher Scientific, USA). Primer sequences are listed in Table 1.

**Table 1. Real-time qPCR primers.**

| Gene | Forward primers | Reverse primers |
|---|---|---|
| iNOS | 5′-TTCCAGAATCCCTGGACAAG-3′ | 5′-TGGTCAAACTCTTGGGGTTC-3′ |
| COX-2 | 5′-AGAAGGAAATGGCTGCAGAA-3′ | 5′-GCTCGGCTTCCAGTATTGAG-3′ |
| TNF-α | 5′-ATGAGCACAGAAAGCATGATC-3′ | 5′-TACAGGCTTGTCACTCGAATT-3′ |
| IL-1β | 5′-GGGCCTCAAAGGAAAGAATC-3′ | 5′-TACCAGTTGGGGAACTCTGC-3′ |
| IL-6 | 5′-AGTTGCCTTCTTGGGACTGA-3′ | 5′-CAGAATTGCCATTGCACAAC-3′ |
| β-actin | 5′-CCACAGCTGAGAGGGAAATC-3′ | 5′-AAGGAAGGCTGGAAAAGAGC-3′ |

## Determination of PGE$_2$, IL-1β, IL-6 and TNF-α

Culture media of cells were collected and centrifuged at 2,000×g for 10 min. The supernatant was used to analyze PGE$_2$ levels with a PGE$_2$ ELISA kit (ADI-900-001, Enzo Life Sciences Inc., USA) according to the manufacturer's instructions. A specific ELISA kit (Abcam, UK) was used to assess concentrations of cytokines, namely IL-1β (ab197742), IL-6 (ab100712), and TNF-α (ab208348).

## Western blot analysis

After treatments, cells were lysed with RIPA buffer (Tech & Innovation, China) for 30 min before centrifuging at 12,000×g for 10 min at 4˚C to prepare cell lysates containing proteins. These cell lysates were used to measure protein concentrations with a Pierce™ BCA Protein Assay Kit (Thermo Fisher Scientific, USA). SDS-PAGE and western blotting were then performed. Primary antibodies against p-NF-κB p65, p-p38, p-JNK, p-ERK1/2 (Cell Signaling Technology, USA) and α-tubulin (Abcam, UK) were used to immunoblot proteins, followed by incubation with goat anti-rabbit IgG(H+L)-HRP (GenDEPOT, USA). Detection of protein signals was conducted using Pierce® ECL Plus Western Blotting Substrate (Thermo Fisher Scientific, USA). A ChemiDoc XRS+ imaging system (Bio-Rad, USA) was then used to capture protein bands.

## Flow cytometry assay

Accessory/ co-stimulatory molecules of cell-surface expression of CD14, CD40, CD86, and TLR4 were identified using flow cytometry. Cells were harvested and washed in a cold-buffer containing 2% FBS and 0.1% sodium azide in phosphate-buffered saline (PBS) before blocking with 50 μL of purified rat IgG (eBioscience Inc., USA) for 10 min. Antibodies against CD40-PE and CD86-APC (eBioscience Inc., USA) were added to the same tubes of cells containing Rat IgG2a kappa-PE and Rat IgG2a kappa-APC, respectively. Antibodies against CD14-FITC and TLR4-PE (eBioscience Inc., USA) were added to the same tubes of cells containing Rat IgG2a kappa-FITC and Rat IgG2a kappa-PE, respectively. All tubes of cells were incubated at 4˚C for an additional 20 min. These cells were rinsed twice with FACS buffer before recording flow cytometry data with a CytoFLEX Flow Cytometer (Beckman Coulter Inc., USA).

## Glycosidic linkage and nuclear magnetic resonance (NMR) analysis

GBP-F1 was analyzed for glycosidic linkage using the approach previously established by Ciucanu and Kerek [39]. A GC-MS system (6890 N/MSD 5973) equipped with an HP-5MS capillary column (30 m × 0.25 mm × 0.25 μm) (Agilent Technologies, USA) was used for the analysis of glycosidic linkage as described previously [40].

To analyze the [1]H NMR spectrum, the most immunobiological polysaccharide GBP-F1 was dissolved in $D_2O$. NMR spectra were generated using a JEOL ECA-600 spectrometer (JEOL, Akishima, Japan) coupled with a 5 mm multinuclear auto-turning TH tunable probe at a base frequency (600 MHz).

### Data analysis

All data were analyzed using IBM SPSS Statistics version 23.0 (SPSS Inc., Chicago, IL, USA). The mean ± standard deviation (SD) of an independent study of three replicates was calculated. Results were examined using one-way analysis of variance (ANOVA) along with Duncan's multiple range tests. A value of $p < 0.05$ was regarded as statistically significant.

## Results

### Cytotoxicity of GBPs to LPS-stimulated RAW264.7 cells

Fig 1 shows a significant increase in cell viability after treatment with LPS alone compared to the control (100%). GBP-C, GBP-F2, and GBP-F3 were not cytotoxic to macrophages at 450 μg/mL. GBP-F1 was not cytotoxic to RAW264.7 cells at concentrations below 250 μg/mL, although it was slightly cytotoxic (92%) at 450 μg/mL. Likewise, ginsenosides isolated from ginseng berries were not cytotoxic to RAW264.7 cells at concentrations between 62.5 and 500 μg/mL [21]. These results revealed that GBPs had no cytotoxicity at concentrations up to 450 μg/mL.

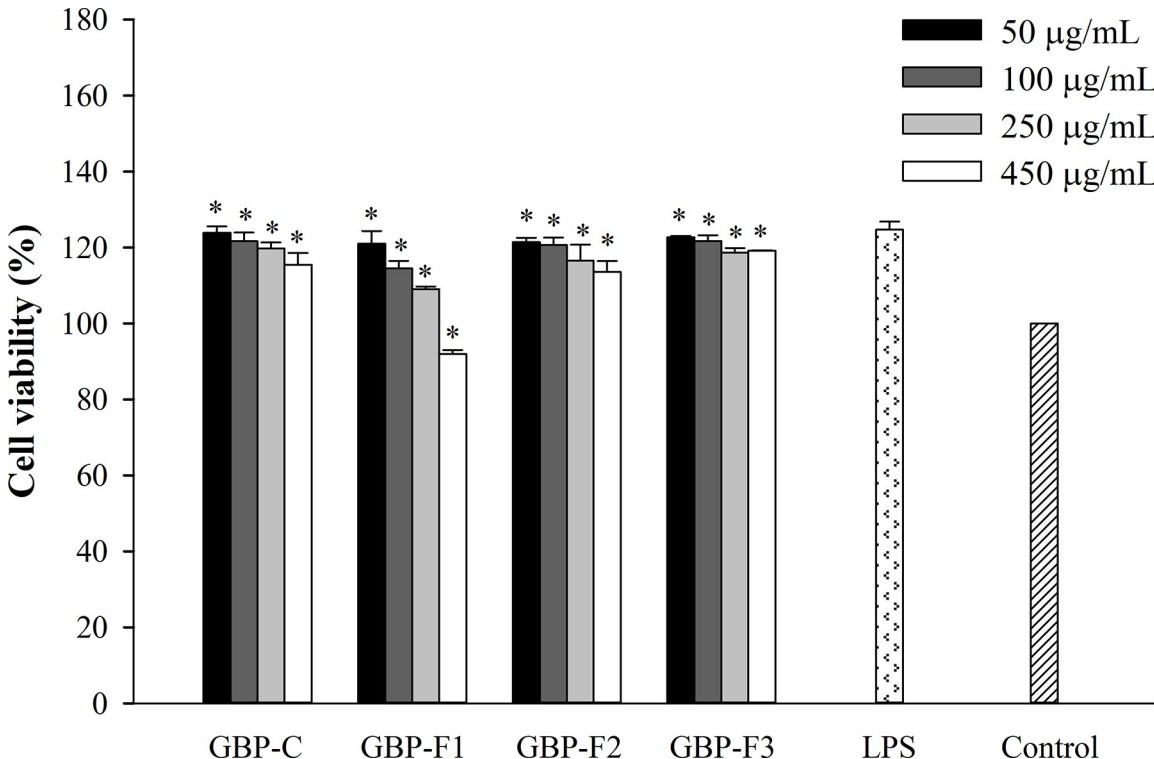

**Fig 1. Cytotoxic effects of GBPs on RAW264.7 cells stimulated by LPS.** Cells were treated with GBPs at various concentrations and stimulated with LPS. Cell viability was assessed using a WST test. Data from three separate experiments performed in duplicate are reported as means ± SD. *$p < 0.05$ vs. RPMI.

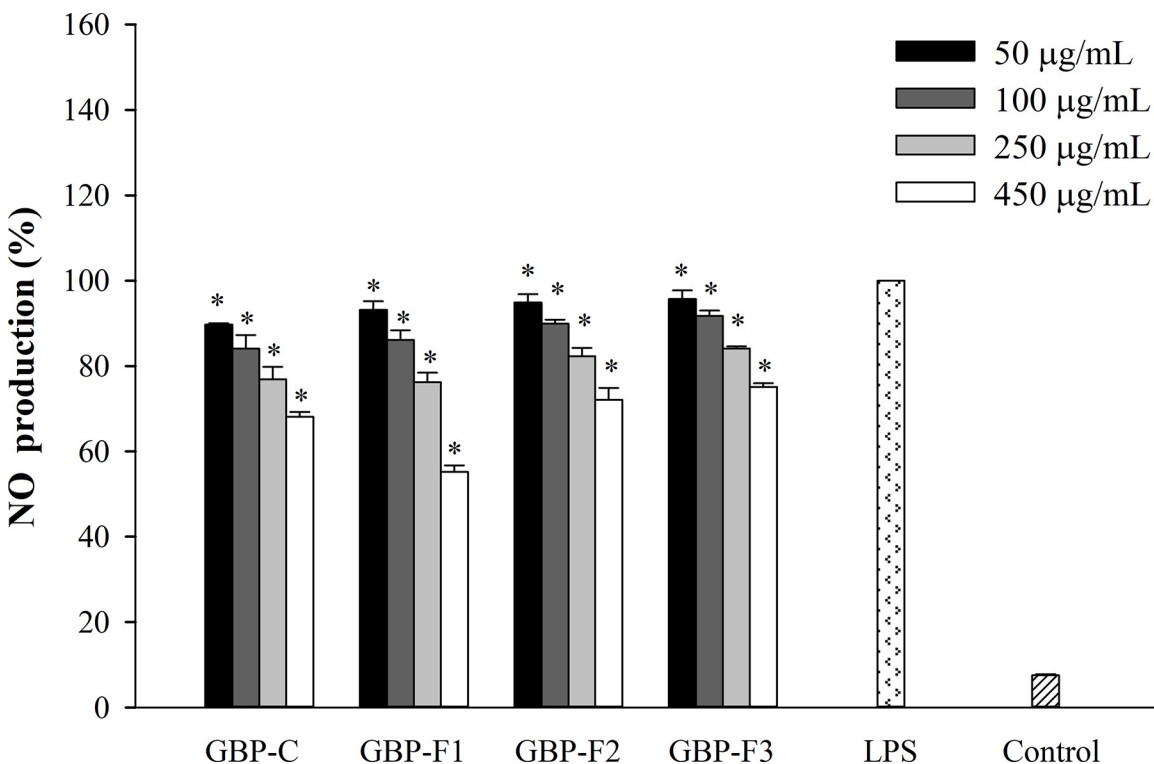

**Fig 2. Effects of GBPs on LPS-stimulated NO production.** Cells were treated with GBPs at various concentrations and stimulated with LPS. The amount of NO was determined with a Griess reagent. Data from three separate experiments performed in duplicate are reported as means ± SD. $^*p < 0.05$ vs. LPS.

## GBPs suppress secretion of inflammatory mediators

Since the production of $PGE_2$ and NO is a significant indicator of inflammation [11, 17, 41], we evaluated effects of GBPs on their production after macrophages were stimulated with LPS. The release of NO production by GBPs in LPS-stimulated RAW246.7 cells was quantified with the Griess reaction method. According to our findings, there were significant dose-related reductions in NO production (50–450 µg/mL) with all GBPs compared to LPS (Fig 2). When LPS-stimulated RAW264.7 cells were treated with GBP-F1 at 450 µg/mL, the highest suppressive effect on NO production was 55%. For subsequent experiments, a concentration of 450 µg/mL was selected for GBPs.

The production of $PGE_2$ was also examined. Prior to stimulation with LPS, cells were pretreated with GBPs at 450 µg/mL. $PGE_2$ concentrations secreted by macrophage cells were then evaluated using an ELISA kit after 24 h of treatment. As shown in Fig 3A, LPS alone considerably boosted $PGE_2$ production in comparison with RPMI, whereas GBPs inhibited levels of $PGE_2$ production similar to those observed with NO production. GBPs also down-regulated the expression of inflammatory mediators such as *iNOS* and *COX-2* when compared to LPS (Fig 3B and 3C). These findings indicated that GBPs suppressed NO and $PGE_2$ production via transcriptional regulation of *iNOS* and *COX-2*, respectively.

## GBPs suppress expression of cytokines

In order to assess effects of GBPs on secretion levels of pro-inflammatory cytokines by LPS-stimulated macrophages, real-time PCR assay and ELISA kits were used to analyze mRNA and

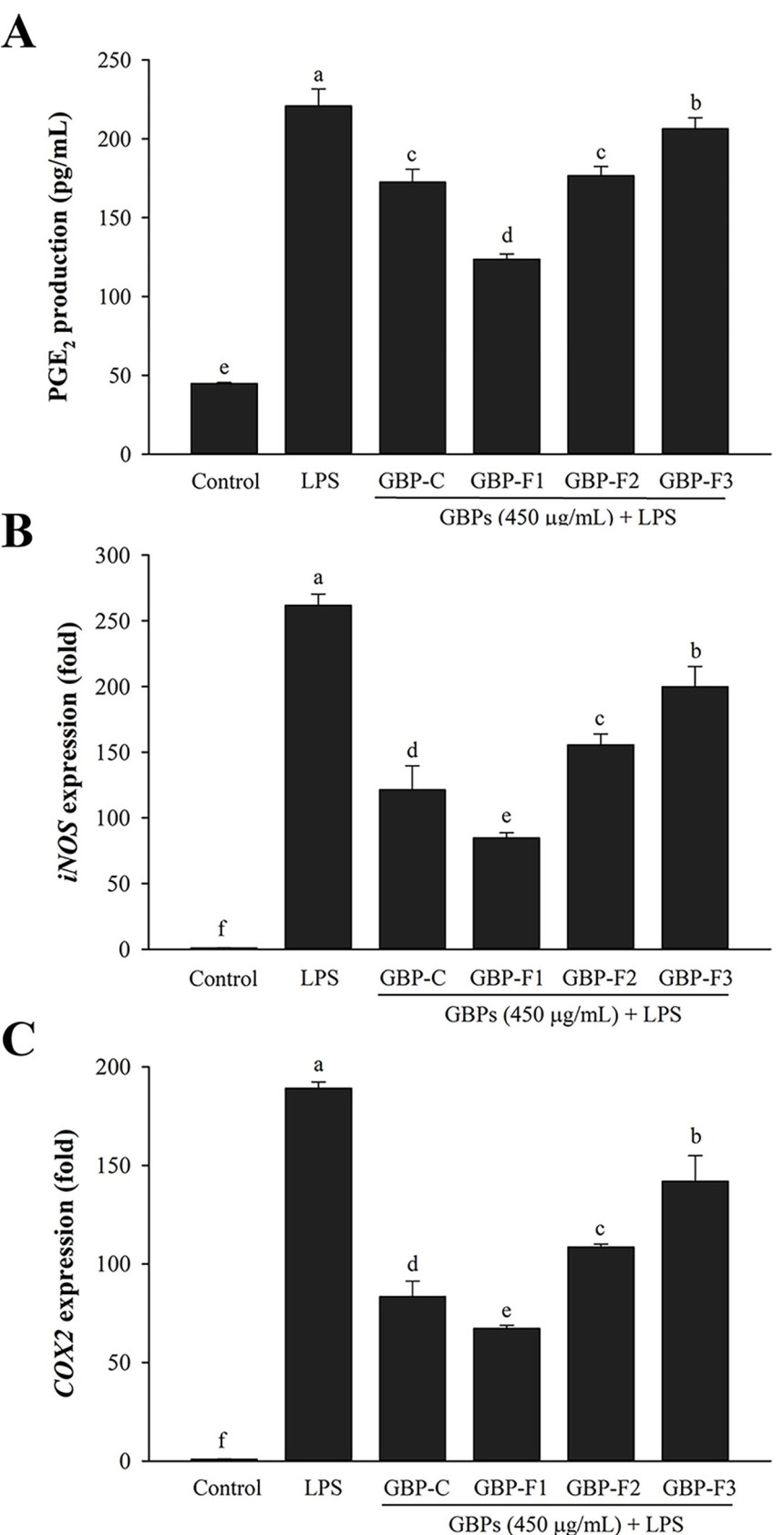

**Fig 3. Effects of GBPs on LPS-stimulated PGE$_2$ production and inflammatory mediators.** Cells were treated with various polysaccharides at 450 μg/mL and stimulated with LPS at 1 μg/mL. Amounts of PGE$_2$ (A) were assessed using ELISA kits. The relative expression of *iNOS* (B) and *COX-2* (C) was assessed using real-time PCR. Data are presented as means ± SD. A different letter ($p < 0.05$) indicates statistically significant differences between treatments.

protein levels, respectively. GBPs reduced levels of *IL-1β*, *IL-6*, and *TNF-α* in RAW264. 7 cells stimulated with LPS (Fig 4A–4C). As shown in Fig 4D–4F, similar results were obtained for protein levels of IL-1β, IL-6, and TNF-α. Thus, GBPs inhibited LPS-induced release of inflammatory cytokines, with GBP-F1 having the highest anti-inflammatory effects. These findings demonstrate that all GBPs, particularly GBP-F1, have potential anti-inflammatory effects.

## GBPs inhibit protein expression of NF-κB and MAPK pathways

Fig 5 shows the anti-inflammatory effects of GBPs on NF-κB and MAPK activation and immune-blotting results. The phosphorylation of NF-κB-p65 was decreased by GBPs in comparison to LPS alone, whereas the phosphorylation of NF-κB-p65 was significantly elevated by GBPs in comparison to the negative control (RPMI). GBPs were also observed in the phosphorylation of ERK, JNK, and p38 in the same pattern of NF-κB activation (Fig 5). Interestingly, GBP-F1 inhibited the phosphorylated NF-κB-p65, ERK, JNK, and p38 in LPS-stimulated RAW264.7 cells at the highest level among the other GBPs. These findings demonstrated that all GBPs, particularly GBP-F1, could suppress the activation of the NF-κB and MAPK pathways.

## GBPs inhibit surface molecule expression

As shown in Fig 6A and 6B, LPS dramatically up-regulated the expression of CD14 and TLR4. However, cell surface expression was considerably decreased by GBPs when compared with

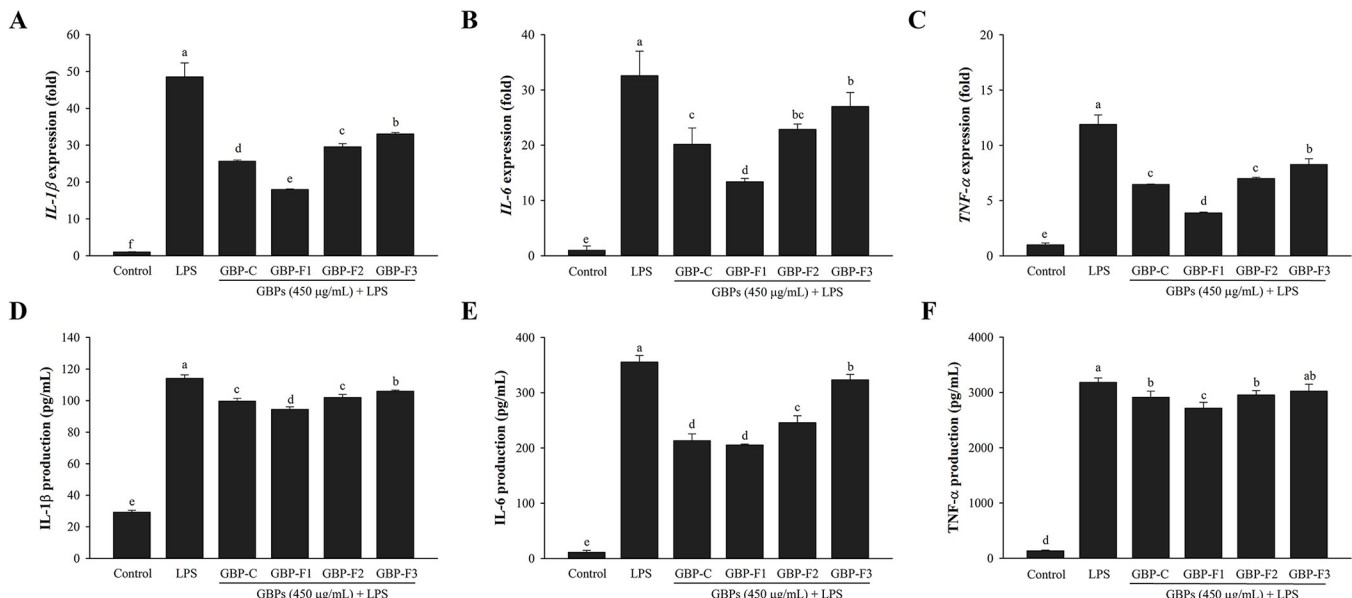

**Fig 4. Effects of GBPs on cytokines released by LPS-stimulated RAW264.7 cells.** Cells were treated with various GBPs at 450 μg/mL and stimulated with LPS at 1 μg/mL. Relative mRNA expression levels of *IL-1β* (A), *IL-6* (B), and *TNF-α* (C) were assessed using real-time PCR. ELISA kits were used to measure protein expression levels of IL-1β (D), IL-6 (E), and TNF-α (F). Data are presented as means ± SD. A different letter ($p < 0.05$) indicates statistically significant differences between treatments.

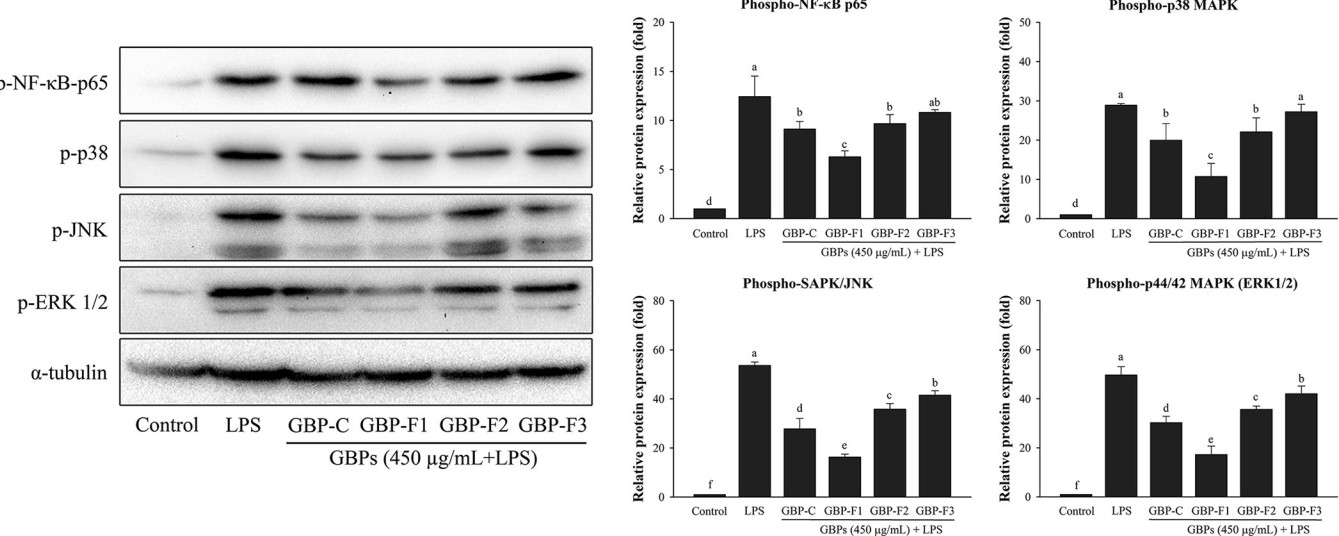

**Fig 5. Effects of GBPs on protein expression by suppressing immune signaling pathways.** Cells were treated with various GBPs at 450 μg/mL and stimulated with LPS at 1 μg/mL. Effects of GBPs on protein expression of NF-κB and MAPK pathways were determined by western blotting. Data from three separate experiments performed in duplicate are reported as means ± SD. A different letter ($p < 0.05$) indicates statistically significant differences between treatments.

LPS alone. GBPs decreased the expression of CD14 and TLR4 in LPS-stimulated macrophages. These findings indicated that GBPs could attenuate LPS binding to the surface of RAW264.7 cells.

In order to examine surface biomarkers of RAW264.7 cells, CD40 and CD86 were subjected to flow cytometry analysis. GBPs were studied for their anti-inflammatory effects on CD40 and CD86 expression. Mean fluorescence intensities (MFI) are shown in Fig 6. GBPs decreased the expression of CD86 but not the expression of CD40 on the surface of LPS-stimulated macrophages (Fig 6C and 6D).

## Glycosidic linkage of polysaccharide from GBP-F1

The strong anti-inflammatory effect of GBP fraction F1 (GBP-F1) was analyzed by GC-MS for glycosidic linkages. As shown in Table 2, ten derivatives were identified: 1→ Ara*f*, 1→ Glc*p*, 1→ Gal*p*, 1→3 Ara*f*, 1→4 Glc*p*, 1→3 Rha*p*, 1→6 Man*p*, 1→6 Gal*p*, 1→3,4 Rha*p*, 1→3,6 Gal*p*. Major derivatives were 1,3,5-tri-O-acetyl-2,4,6-tri-O-methyl-Ara (22.7 ± 0.6%) and 1,4,5-tri-O-acetyl-2,3,6-tri-O-methyl-Glc, (21.3 ± 0.1%), suggesting that GBP-F1 was mainly composed of 1,3-linked arabinofuranose and 1,4-linked glucopyranose linkages. Linkages residues also included 1,5,6-tri-O-acetyl-2,3,4-tri-O-methyl-Gal, 1,3,5-tri-O-acetyl-2,4,6-tri-O-methyl-Rha, and 1,5,6-tri-O-acetyl-2,3.4-tri-O-methyl-Man, implying connections at 1,6-linked galactopyranose, 1,3-linked rhamnopyranose, and 1,6-linked manopyranose. In addition, residues 1,3,4,5-tretra-O-acetyl-2,6-di-O-methyl-Rha, and 1,3,5,6-tretra-O-acetyl-2,4-di-O-methyl-Glc showed some branches of 1,3,4-linked rhamnopyranose and 1,3,6-linked galactopyranose, implying that these branches might be connected to main residues at O-4 rhamnose and O-3 galactose.

Glycosidic linkages of GBP-F1 polymer were further identified from 1D NMR spectra (Fig 7) with $^1$H analysis to support the ten derivatives of glycosidic linkages residues. All ten anomeric signals were noticed from $^1$H spectra, showing signals at 5.48, 5.40, 5.31, 5.28, 5.08, 4.83, 4.72, 4.56, 4.42, and 4.23, respectively.

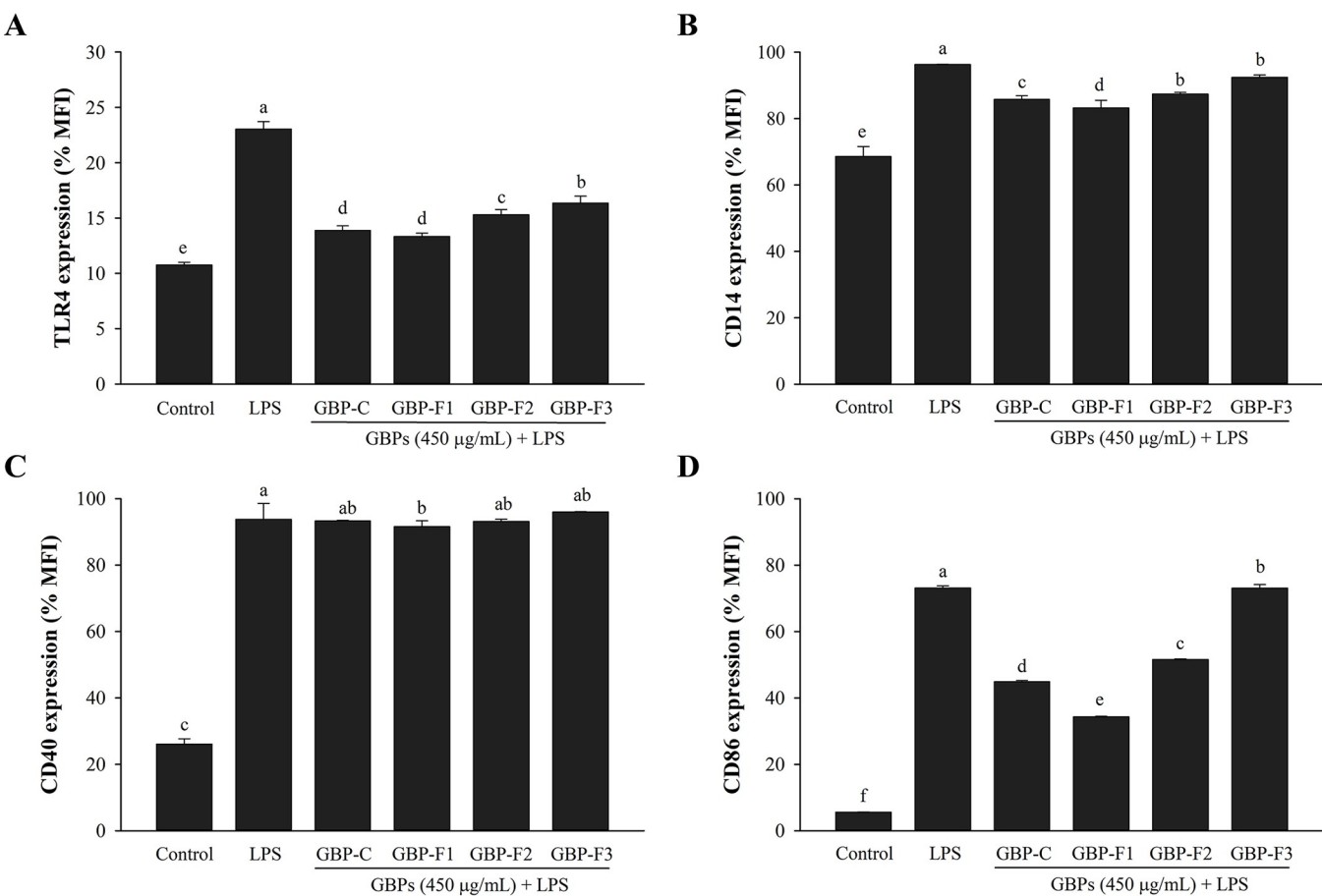

**Fig 6. Effects of GBPs on LPS-induced surface molecule expression.** Cells were treated with various GBPs at 450 µg/mL and stimulated with LPS at 1 µg/mL. Effects of GBPs on expression of cell surface molecules of CD14 (A), TLR4 (B), CD40 (C), and CD86 (D) were then determined. Data are presented as means ± SD. A different letter ($p < 0.05$) indicates statistically significant differences between treatments.

## Discussion

In our preliminary research study, we have isolated, purified, and characterized a crude polysaccharide and fractionated polysaccharides (GBP-C, GBP-F1, GBP-F2, and GBP-F3) and further confirmed that these four polysaccharides consist of five monosaccharides, including

**Table 2. Methylation analysis of GBP-F1.**

| Reaction time (min) | Methylation | Glycosidic linkage | Peak area (%) |
|---|---|---|---|
| 5.766 | 1,4-di-O-acetyl-2,3,5-tri-O-methyl-Ara | Ara*f*-(1→ | 3.3±0.2 |
| 8.461 | 1,5-di-O-acetyl-2,3,4,6-tetra-O-methyl-Glc | Glc*p*-(1→ | 5.6±0.4 |
| 8.815 | 1,5-di-O-acetyl-2,3,4,6-tetra-O-methyl-Gal | Gal*p*-(1→ | 7.2±0.7 |
| 10.079 | 1,3,5-tri-O-acetyl-2,4,6-tri-O-methyl-Ara | →3)-Ara*f*-(1→ | 22.7±0.6 |
| 10.217 | 1,4,5-tri-O-acetyl-2,3,6-tri-O-methyl-Glc | →4)-Glc*p*-(1→ | 21.3±0.1 |
| 10.363 | 1,3,5-tri-O-acetyl-2,4,6-tri-O-methyl-Rha | →3)-Rha*p*-(1→ | 7.7±0.5 |
| 10.534 | 1,5,6-tri-O-acetyl-2,3.4-tri-O-methyl-Man | →6)-Man*p*-(1→ | 4.5±0.3 |
| 11.071 | 1,5,6-tri-O-acetyl-2,3,4-tri-O-methyl-Gal | →6)-Gal*p*-(1→ | 10.4±0.2 |
| 11.288 | 1,3,4,5-tretra-O-acetyl-2,6-di-O-methyl-Rha | →3,4)-Rha*p*-(1→ | 8.7±0.4 |
| 12.693 | 1,3,5,6-tretra-O-acetyl-2,4-di-O-methyl-Gal | →3,6)-Gal*p*-(1→ | 7.0±0.5 |

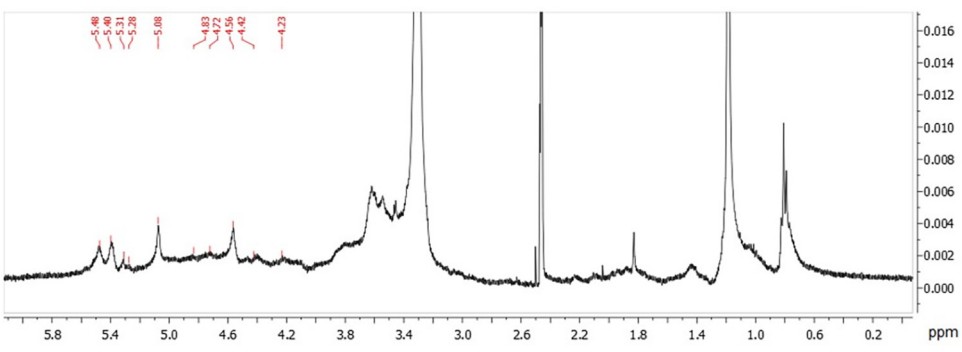

**Fig 7. ¹H spectrum of GBP-F1.**

rhamnose (4.0–15.9%), arabinose (10.7–26.8%), mannose (2.2–5.9%), glucose (26.7–34.6%), and galactose (24.5–46.3%) [32]. These four polysaccharides have also been demonstrated to improve immunological function of macrophages. However, their anti-inflammatory effects have not been reported yet. In this study, anti-inflammatory effects of Korean ginseng berry polysaccharides on macrophages were investigated. The mechanism by which these four polysaccharides suppressed LPS-induced inflammatory responses was also determined.

Macrophages play an essential role in both particular and non-specific immunological reactions throughout the inflammatory process by generating a variety of inflammation-related bioactive chemicals, mediators, and cytokines, most notably NO, $PGE_2$, TNF-α, and several specific interleukins [2, 9]. NO synthesized from arginine by nitric oxide synthase (NOS) is an important mediator of inflammation in the body since it is a neurotransmitter and a defense molecule against bacteria, parasites, and tumor cells [9]. In the present study, results indicated that NO generation was decreased in groups treated with GBPs at 50–450 μg/mL. A high inhibition on NO production was found in the GBP-F1 group. These results might be due to expression of *iNOS* induced by GBPs in LPS-stimulated RAW264.7 cells. Saponins extracted from ginseng and sulfated polysaccharides from brown alga can also effectively suppress LPS-induced NO production up to 400 μg/mL of the concentration of saponin extracts [10, 14]. $PGE_2$ is produced at the inflammatory site by COX-2. It is also implicated as an important mediator in the process of inflammation. Thus, reducing *iNOS* and *COX-2* expression in macrophages may be an efficient trigger for inhibiting the production of NO and $PGE_2$ to suppress inflammation. Our results showed that GBPs at 450 μg/mL also reduced $PGE_2$ production and mRNA expression levels of *iNOS* and *COX-2*. Similarly, guluronate oligosaccharides from alginate also decreased expression levels of mRNA and proteins of iNOS and COX-2 in LPS-stimulated macrophages [41]. These findings suggest that GBPs can decrease NO and $PEG_2$ production by decreasing *iNOS* and *COX-2* expression, thus exerting potent anti-inflammatory effects. Polysaccharides from *Smilax glabra* Roxb., such as SGP-1 and SGP-2, have been shown to be able to reduce *iNOS*, *TNF-α*, and *IL-6* expression in LPS-stimulated macrophages [16]. TNF-α, IL-6, and IL-1β are primary cytokines generated by activated macrophages during initial inflammatory responses of host defense and infection [2], consistent with previous reports [10, 21]. In this study, GBPs dramatically reduced the synthesis of IL-1β, IL-6, and TNF-α in LPS-stimulated RAW264.7 cells, indicating that GBPs could inhibit macrophage immune responses by directly increasing cytokine production and mRNA expression.

Recently, numerous studies have discovered that polysaccharides produced from medicinal plants possess anti-inflammatory activities by blocking activation of NF-κB and MAPK signaling pathways in macrophages [16–18]. Transcription factor NF-κB is important for regulating

inflammation-induced enzymes and cytokines in a variety of cellular functions, including NF-κB1 (p50/p105), NF-κB2 (p52/p100), p65 (RelA), RelB, and c-Rel, which are recognized as biomarkers of NF-κB activation [42]. Activated NF-κB p65 can migrate to the nucleus and regulate inflammation-related genes such as iNOS and pro-inflammatory cytokines [10]. Interestingly, our results revealed that GBPs reduced NF-κB release in LPS-stimulated RAW264.7 cells by reducing NF-κB phosphorylation and translocation to the nucleus. The MAPK pathway is also a key intracellular signaling cascade in immune responses. MAPK has three families, namely, p38 MAPK, JNK/stress activated protein kinase, and ERK [43]. Our results also demonstrated that GBPs suppressed ERK1/2 and p38 phosphorylation. These findings imply that anti-inflammatory actions of GBPs via NF-κB signaling regulation might contribute to their inhibitory effects on MAPK activity.

LPS can stimulate macrophages to release inflammatory mediators related to TLR4 and CD14 molecules on the cell surface that play important roles in the immune system [44]. LPS signaling appears when LPS is transferred from membrane to soluble CD14 by LPS binding protein, which is subsequently recognized by TLR4 [5, 13]. Guluronate oligosaccharides from alginate can decrease LPS-stimulated expression of TLR4 and CD14 molecules [41]. Polysaccharides from thunder god vine (*Tripterygium wilfordii* Hook. f.) can also reduce the release of adhesion molecules including CD11c, CD18, CD14, and CD54 in THP-1 cells [45]. Co-stimulatory molecules such as CD40 and CD86 are expressed on APCs and T cells, indicating active macrophages on their surfaces [5, 11, 18]. In the present study, results indicated that CD86 expression decreased the immunological activity in RAW264.7 cells treated with GBPs. Reduced expression of CD40 and CD86 can alleviate diseases such as chronic inflammatory disease and autoimmune disease because they are crucial for productive interactions between T cells and APCs cells [5]. These results suggest that surface molecules of CD14, CD86, and TLR4 might contribute to inflammatory responses of LPS-stimulated RAW264.7 macrophages treated by GBPs.

In addition to chemical composition, molecular weight, conformation, and other structure-activity relationships, polysaccharides can be evaluated according to their functional anti-inflammatory properties [46]. GBP-F1 contained rhamnose (14.1%), arabinose (26.8%), mannose (4.4%), glucose (26.7%), and galactose (24.5%) as monossacharide components, with molecular weight of $115.7 \times 10^3$ g/mol [32]. The higher activity of GBP-F1 included higher amount of (1→3)-linked Ara*f*, (1→4)-linked Glc*p*, (1→6)-linked Gal*p* residues as the main backbone of GBP-F1. The connection type of glycosidic linkages presumably affects polysaccharides' anti-inflammatory activities. A previous study has found that fucomannogalactan isolated from *Amanita muscaria* that contains (1 → 3) and (1 → 6) linked α-D-Glc*p* presents antinociceptive and anti-inflammatory properties [47]. Similarly, exopolysaccharides from *Crypthecodinium cohnii* also consist of (1→6)-linked Man*p* and (1→6)-linked Gal*p* residues with (1→3,6)-linked Man*p*. They also exhibit anti-inflammatory activities [8]. NMR results corresponded with glycosidic linkage analysis results. NMR is a typical tool for characterizing structural characteristics of polysaccharides. Sugar residues are assigned $^1$H NMR spectrum and reference data [47–49]. In this study, $^1$H NMR spectra of GBP-F1 showed a significant correlation in the anomeric region between 5.40 and 5.08 ppm [50]. The backbone structure of GBP-F1 was a little different from the polysaccharide of *Dendrobium huoshanense*, which comprised (1→6)-linked β-D-Glc*p*, (1→4)-linked β -D-Glc*p*, and (1,4→6)-linked β–D-Glc*p* [51]. Moreover, glycosidic linkages of polysaccharide from American ginseng and *Laminaria japonica* were markedly varied, showing different degrees of branching and diverse monosaccharide compositions [48, 49]. These findings suggest that GBP-F1 might be responsible for anti-inflammatory activity. Moreover, anti-inflammatory activities of polysaccharides are intricately linked to their chemical compositions and molecular characteristics such as molecular

weight, monosaccharide composition, and glycosidic linkage type. Nevertheless, further research is needed to elucidate the precise mechanism underlying the relationship between activity and structure.

## Conclusions

In summary, anti-inflammatory effects of all GBPs, including fractionated GBP-C, GBP-F1, GBP-F2, and GBP-F3, are mediated by inhibiting inflammatory mediators and cytokines as well as surface accessory/co-stimulatory molecules via activation of NF-κB and MAPK signaling pathways in LPS-stimulated RAW264.7 cells. In particular, GBP-F1 showed more potent anti-inflammation than GBP-C, GBP-F2, and GBP-F3 fractions. GBP-F1 demonstrated significant anti-inflammatory properties. It has the potential to be a natural product used in the development of therapeutic agents.

## Supporting information

**S1 Fig. Original western blot gel image data.**
(PDF)

## Author Contributions

**Conceptualization:** Woo Jung Park.

**Data curation:** Woo Jung Park.

**Formal analysis:** Weerawan Rod-in, Utoomporn Surayot.

**Funding acquisition:** Woo Jung Park.

**Investigation:** Weerawan Rod-in.

**Methodology:** Weerawan Rod-in.

**Project administration:** Woo Jung Park.

**Resources:** Woo Jung Park.

**Software:** Weerawan Rod-in.

**Supervision:** Woo Jung Park.

**Validation:** Weerawan Rod-in.

**Visualization:** Weerawan Rod-in.

**Writing – original draft:** Weerawan Rod-in.

**Writing – review & editing:** Utoomporn Surayot, SangGuan You, Woo Jung Park.

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
