## [Decision Letter · Decision Letter 0]

22 Aug 2023

PONE-D-23-20233Inhibitory effects of polysaccharides from Korean ginseng berries on LPS-induced RAW264.7 macrophagesPLOS ONE

Dear Dr. Park,

Thank you for submitting your manuscript to PLOS ONE. After careful consideration, we feel that it has merit but does not fully meet PLOS ONE’s publication criteria as it currently stands. Therefore, we invite you to submit a revised version of the manuscript that addresses the points raised during the review process.

We look forward to receiving your revised manuscript.

Kind regards,

Hanbing Li, Ph.D.

Academic Editor

PLOS ONE

Journal Requirements:

Reviewers' comments:

Reviewer's Responses to Questions

**Comments to the Author**

1. Is the manuscript technically sound, and do the data support the conclusions?

Reviewer #1: Yes

Reviewer #2: Partly

2. Has the statistical analysis been performed appropriately and rigorously? 

Reviewer #1: N/A

Reviewer #2: Yes

3. Have the authors made all data underlying the findings in their manuscript fully available?

Reviewer #1: Yes

Reviewer #2: Yes

4. Is the manuscript presented in an intelligible fashion and written in standard English?

Reviewer #1: Yes

Reviewer #2: Yes

5. Review Comments to the Author

Reviewer #1: It is a well written manuscript justifying the title of the manuscript. An extensive work has been undertaken by the Scientist. The study carried out demonstrate that GBPs can be employed as an alternative natural

source of anti-inflammatory agents

Reviewer #2: In a manuscript entitled „Inhibitory effects of polysaccharides from Korean ginseng berries on LPS-induced RAW264.7 macrophages“ authors show the effect of GBPs on the functional response of macrophages in the RAW264.7 murine cell line. The manuscript is standardly structured, but the individual sections need editing to bring them to publication quality. Mainly the English needs to be improved, especially the sentence structure - subject and object. I will give some examples below.

I have the following questions and comments about the manuscript:

- In the introduction, you are the first to emphasize macrophages and their activation. But it is not clear in the discussion or the conclusion of the manuscript why you chose macrophages. What is the aim of this study? What is the possible application/use of these polysaccharides in relation to macrophages and possible treatment?

- In my opinion, the introduction should be improved. At a minimum, highlight the novelty of the study and why it is necessary to publish these results.

- The materials and methods section lacks a cell culture section. It is not clear how they were cultured, whether RMPI is a control with no added substances or extra RPMI medium was added, etc.

- From my own experience and that of other laboratories, the 1ug/ml LPS concentration is extremely high. The standard used is 10-100ng/ml max. 200 ng/ml. This high concentration is indicative of a poor state of the cells, which are unable to be activated by lower concentrations. Or you are overactivating your cells unnecessarily and losing the effective effect of the substances.

- I assume that the treatment and activation of GBPs and LPS cells is still the same for all methods. Therefore, I would recommend that you can describe one treatment for the first method and then just write that the medium/cells will be used for further analyses or keep the treatment sentences the same. Now as you are trying to have different sentences it kind of evokes a different treatment procedure.

- For WB analysis, activation times and exact sample preparation procedure need to be written. Because both the time of the activation is not there now and the time of the sample preparation is important for protein phosphorylation. Was there any pretreatment of GBPs for protein phosphorylation?

- line 128-9 you wrote: „The antibodies against CD14-FITC, CD40-PE, CD86-APC, and TLR4-PE (eBioscience Inc, USA). were added to the appropriate tubes together with their isotype control“ Do you put a CD marker in one tube together with an iso control? How do you distinguish on the flow analyzer what is the signal from the ISO control and what is the signal from the CD marker? Do you use ISO controls for compensation? How much did you have to compensate for the FITC and PE signal?

- In the results section there are sentences from the methodologies, since you have already described it once in the methodology, there is no need to repeat it again. For example, the sentence line 149: „The toxicity of LPS-stimulated RAW264.7 cells was examined using the EZ-Cytox Cell Viability Assay Kit and various concentrat“.

- I would recommend rewriting RPMI to control in the images, because I assume the RPMI medium is in all samples.

- Throughout the text you use abbreviations like GBP-C, GBP-F1... so correct that in the Figures accordingly.

- line 232, you wrote: „However, the cell surface expression of GBPs was considerably decreased in the only LPS treatment group.“ Have you detected GBPs on the cell surface? If yes, please state in the methodology and results.

- You selected GBP-F1 as the best, but if all results were normalised to cytotoxicity it would not be so significant. Because that fraction has the highest cytotoxicity. the lowest concentration has about 120% and the highest has 92% which is almost 30% cytotoxicity. Why don't you also mention LPS control for cytotoxicity? How do you explain that lower concentrations of GBPs increase the percentage in cytotoxicity tests?

- The discussion section is more of an results with the introduction than the discussion itself. This part of the manuscript needs to be rewritten and improved with a deeper discussion of the results achieved.

- The conclusion needs to be rewritten. it is necessary that the conclusion is evident from it, not just a summary of the results.

Overall, this manuscript did not convince me that it is suitable for publication in its current quality. Since there are a lot of things to redo and especially to clarify the novelty and the goal of the work, that's why I'm giving now REJECT.

6. PLOS authors have the option to publish the peer review history of their article (what does this mean?). If published, this will include your full peer review and any attached files.

Reviewer #1: **Yes: **Dr.Harsha Kharkwal FRSC

Reviewer #2: No

---

## [Author Response · Author response to Decision Letter 0]

17 Sep 2023

Reviewer #1: It is a well written manuscript justifying the title of the manuscript. An extensive work has been undertaken by the Scientist. The study carried out demonstrate that GBPs can be employed as an alternative natural source of anti-inflammatory agents

Response: Thanks.

Reviewer #2: In a manuscript entitled „Inhibitory effects of polysaccharides from Korean ginseng berries on LPS-induced RAW264.7 macrophages“ authors show the effect of GBPs on the functional response of macrophages in the RAW264.7 murine cell line. The manuscript is standardly structured, but the individual sections need editing to bring them to publication quality. Mainly the English needs to be improved, especially the sentence structure - subject and object. I will give some examples below.

I have the following questions and comments about the manuscript:

- In the introduction, you are the first to emphasize macrophages and their activation. But it is not clear in the discussion or the conclusion of the manuscript why you chose macrophages. What is the aim of this study? What is the possible application/use of these polysaccharides in relation to macrophages and possible treatment?

Response: It was corrected. Thanks.

- In my opinion, the introduction should be improved. At a minimum, highlight the novelty of the study and why it is necessary to publish these results.

Response: It was corrected. Thanks.

- The materials and methods section lacks a cell culture section. It is not clear how they were cultured, whether RMPI is a control with no added substances or extra RPMI medium was added, etc.

Response: It was corrected in methodology. Thanks.

- From my own experience and that of other laboratories, the 1 ug/ml LPS concentration is extremely high. The standard used is 10-100 ng/ml max. 200 ng/ml. This high concentration is indicative of a poor state of the cells, which are unable to be activated by lower concentrations. Or you are over activating your cells unnecessarily and losing the effective effect of the substances.

Response: In this experiment, we were used the final concentration of LPS at 1 μg/mL. Thanks.

• Many studied associated with anti-inflammatory activity of polysaccharides were used LPS at 1 μg/ml LPS.

• References

• Xiong, Q., Hao, H., He, L., Jing, Y., Xu, T., Chen, J., Zhang, H., Hu, T., Zhang, Q., Yang, X. and Yuan, J., 2017. Anti-inflammatory and anti-angiogenic activities of a purified polysaccharide from flesh of Cipangopaludina chinensis. Carbohydrate polymers, 176, pp.152-159.

• Wu, G.J., Shiu, S.M., Hsieh, M.C. and Tsai, G.J., 2016. Anti-inflammatory activity of a sulfated polysaccharide from the brown alga Sargassum cristaefolium. Food Hydrocolloids, 53, pp.16-23.

• Hwang, P.A., Chien, S.Y., Chan, Y.L., Lu, M.K., Wu, C.H., Kong, Z.L. and Wu, C.J., 2011. Inhibition of lipopolysaccharide (LPS)-induced inflammatory responses by Sargassum hemiphyllum sulfated polysaccharide extract in RAW 264.7 macrophage cells. Journal of agricultural and food chemistry, 59(5), pp.2062-2068.

• Bhardwaj, M., Padmavathy, T.K., Mani, S., Malarvizhi, R., Sali, V.K. and Vasanthi, H.R., 2020. Sulfated polysaccharide from Turbinaria ornata suppress lipopolysaccharide-induced inflammatory response in RAW 264.7 macrophages. International journal of biological macromolecules, 164, pp.4299-4305.

• Zhou, R., Shi, X., Gao, Y., Cai, N., Jiang, Z. and Xu, X., 2015. Anti-inflammatory activity of guluronate oligosaccharides obtained by oxidative degradation from alginate in lipopolysaccharide-activated murine macrophage RAW 264.7 cells. Journal of agricultural and food chemistry, 63(1), pp.160-168.

• Zhou, R., Shi, X., Gao, Y., Cai, N., Jiang, Z. and Xu, X., 2015. Anti-inflammatory activity of guluronate oligosaccharides obtained by oxidative degradation from alginate in lipopolysaccharide-activated murine macrophage RAW 264.7 cells. Journal of agricultural and food chemistry, 63(1), pp.160-168.

- I assume that the treatment and activation of GBPs and LPS cells is still the same for all methods. Therefore, I would recommend that you can describe one treatment for the first method and then just write that the medium/cells will be used for further analyses or keep the treatment sentences the same. Now as you are trying to have different sentences it kind of evokes a different treatment procedure.

Response: It was corrected in methodology. Thanks.

- For WB analysis, activation times and exact sample preparation procedure need to be written. Because both the time of the activation is not there now and the time of the sample preparation is important for protein phosphorylation. Was there any pretreatment of GBPs for protein phosphorylation?

Response: It was corrected in methodology. Thanks.

- line 128-9 you wrote: „The antibodies against CD14-FITC, CD40-PE, CD86-APC, and TLR4-PE (eBioscience Inc, USA). were added to the appropriate tubes together with their isotype control“ Do you put a CD marker in one tube together with an iso control? How do you distinguish on the flow analyzer what is the signal from the ISO control and what is the signal from the CD marker? Do you use ISO controls for compensation? How much did you have to compensate for the FITC and PE signal?

Response: It was corrected in methodology. Thanks.

• The antibodies against CD40-PE and CD86-APC (eBioscience Inc., USA) were added to the same tubes of treated-sample cells with Rat IgG2a kappa-PE and Rat IgG2a kappa-APC, respectively. The antibodies against CD14-FITC and TLR4-PE (eBioscience Inc., USA) were added to the same tubes of treated-sample cells with Rat IgG2a kappa-FITC and Rat IgG2a kappa-PE, respectively.

• Fluorescent channel of each CytoFLEX channel different

o Fluorescent channel of 525/40 BP for CytoFLEX channel – FITC

o Fluorescent channel of 585/42 BP for CytoFLEX channel – PE

o Fluorescent channel of 660/10 BP for CytoFLEX channel – APC

- In the results section there are sentences from the methodologies, since you have already described it once in the methodology, there is no need to repeat it again. For example, the sentence line 149: „The toxicity of LPS-stimulated RAW264.7 cells was examined using the EZ-Cytox Cell Viability Assay Kit and various concentrations “.

Response: It was corrected in methodology. Thanks.

- I would recommend rewriting RPMI to control in the images, because I assume the RPMI medium is in all samples.

Response: It was corrected. Thanks.

- Throughout the text you use abbreviations like GBP-C, GBP-F1... so correct that in the Figures accordingly.

Response: It was corrected. Thanks.

- line 232, you wrote: „However, the cell surface expression of GBPs was considerably decreased in the only LPS treatment group.“ Have you detected GBPs on the cell surface? If yes, please state in the methodology and results.

Response: It was corrected. Thanks.

- You selected GBP-F1 as the best, but if all results were normalised to cytotoxicity it would not be so significant. Because that fraction has the highest cytotoxicity. the lowest concentration has about 120% and the highest has 92% which is almost 30% cytotoxicity. Why don't you also mention LPS control for cytotoxicity? How do you explain that lower concentrations of GBPs increase the percentage in cytotoxicity tests?

Response: It was corrected. Thanks.

• In our study, the four fractions of polysaccharides showed the highest NO production at a concentration of 450 μg/mL, especially GBP-F1, which reduced NO production by 55% while causing no toxic in the cells. All experiments with four polysaccharides should use the same concentration.

• We changed Figure 1, which added the cytotoxicity of LPS and control (RPMI).

- The discussion section is more of an results with the introduction than the discussion itself. This part of the manuscript needs to be rewritten and improved with a deeper discussion of the results achieved.

Response: It was corrected. Thanks.

- The conclusion needs to be rewritten. it is necessary that the conclusion is evident from it, not just a summary of the results.

Response: It was corrected. Thanks.

Overall, this manuscript did not convince me that it is suitable for publication in its current quality. Since there are a lot of things to redo and especially to clarify the novelty and the goal of the work, that's why I'm giving now REJECT.

Response: According to your suggestion, the manuscript was corrected. Please check it again and we are looking forward to hearing from you. Thanks.

---

## [Decision Letter · Decision Letter 1]

6 Nov 2023

Inhibitory effects of polysaccharides from Korean ginseng berries on LPS-induced RAW264.7 macrophages

PONE-D-23-20233R1

Dear Dr. Woo Jung Park,

We’re pleased to inform you that your manuscript has been judged scientifically suitable for publication and will be formally accepted for publication once it meets all outstanding technical requirements.

Kind regards,

Hanbing Li, Ph.D.

Academic Editor

PLOS ONE

Additional Editor Comments (optional):

The comments and questions raised by the reviewers have been addressed, while some minor issues from the reviewers should be improved before the manuscript is completely accepted and published.

Reviewers' comments:

Reviewer's Responses to Questions

**Comments to the Author**

1. If the authors have adequately addressed your comments raised in a previous round of review and you feel that this manuscript is now acceptable for publication, you may indicate that here to bypass the “Comments to the Author” section, enter your conflict of interest statement in the “Confidential to Editor” section, and submit your "Accept" recommendation.

Reviewer #2: All comments have been addressed

Reviewer #3: All comments have been addressed

2. Is the manuscript technically sound, and do the data support the conclusions?

Reviewer #2: (No Response)

Reviewer #3: Yes

3. Has the statistical analysis been performed appropriately and rigorously? 

Reviewer #2: Yes

Reviewer #3: Yes

4. Have the authors made all data underlying the findings in their manuscript fully available?

Reviewer #2: Yes

Reviewer #3: Yes

5. Is the manuscript presented in an intelligible fashion and written in standard English?

Reviewer #2: Yes

Reviewer #3: Yes

6. Review Comments to the Author

Reviewer #2: (No Response)

Reviewer #3: The author has addressed all the questions raised by reviewers. As of now, the manuscript appears to be in good shape and is suitable for publication in this journal. However, before proceeding with publication, the author should address the following issues:

1. Improve the abstract section.

2. Recheck the grammar and proofread for typos in the manuscript.

3. Enhance the quality of all figures.

7. PLOS authors have the option to publish the peer review history of their article (what does this mean?). If published, this will include your full peer review and any attached files.

Reviewer #2: No

Reviewer #3: **Yes: **Narayanasamy Marimuthu Prabhu

---

## [Editor Report · Acceptance letter]

16 Nov 2023

PONE-D-23-20233R1 

Inhibitory effects of polysaccharides from Korean ginseng berries on LPS-induced RAW264.7 macrophages 

Dear Dr. Park:

I'm pleased to inform you that your manuscript has been deemed suitable for publication in PLOS ONE. Congratulations! Your manuscript is now with our production department. 

Kind regards, 

on behalf of

Professor Hanbing Li 

Academic Editor

PLOS ONE